# Fathers’ Educational Needs Assessment in Relation to Their Participation in Perinatal Care: A Systematic Review

**DOI:** 10.3390/healthcare11020200

**Published:** 2023-01-09

**Authors:** Zoi Palioura, Antigoni Sarantaki, Evangelia Antoniou, Maria Iliadou, Maria Dagla

**Affiliations:** Department of Midwifery, School of Health & Care Sciences, University of West Attica, 12243 Athens, Greece

**Keywords:** fathers, antenatal education, prenatal care and perinatal care

## Abstract

Even though they are crucial for a positive pregnancy experience, many fathers may not be aware of the significance of their role in perinatal care. As it is important to identify the needs of the target population in the initial phases of a health plan in order to ensure success, the current systematic review is the first one to address the reported needs for men’s antenatal education. Methods: All studies written in English and published between 1997 and 2021 relevant to the theme were included in the review. The electronic databases of various mainstream journals were used to evaluate 87 studies. Results: In total, the validity of 17 studies was determined based on their compliance with the inclusion criteria. According to the review, men’s participation in antenatal education can significantly influence pregnancy, childbirth, parenting, couple’s relationship, and overall family functioning. Conclusion: Providing face-to-face antenatal education to men by doctors or midwives is emphasized as an important component of perinatal care preparation because it leads to positive outcomes.

## 1. Introduction

One-half of the active population consists of men, who make decisions about health, education, economic activities of their partners, and family matters in general [1,2]. Innately, men’s participation in women’s care is one of the key strategies to achieving the third Millennium Development Goals of empowering women and improving maternal health [3]. The World Health Organization’s Recommendations on Health Promotion Interventions for Maternal and Newborn Health recommends interventions to engage fathers during pregnancy, childbirth, and the postnatal period. The promotion of men’s participation in safe motherhood programs is crucial to increasing awareness and participation in childbirth planning, such as facilitating access to and use of perinatal care [4]. Men are willing to learn more about reproductive health and its various aspects, according to some studies. [5,6]. A further concern is identifying appropriate strategies for men’s involvement based on an assessment of their needs [7,8].

Training is essential for changing and improving the beliefs and knowledge of men, and it contributes to their role as partners and fathers [7]. A standardized, step-by-step training process begins with identifying educational needs [9]. In addition, educational programs can create a framework that aims to address other important components of priority needs. Both can provide the groundwork for defining the objectives and the appropriate framework for addressing additional significant priority factors [10]. The best method to categorize educational needs is by asking learners [11].

When expectant fathers are involved in antenatal education, health behavior during pregnancy can be improved [12], and women’s use of skilled childbirth and postpartum care can be increased [13], as well as the presence of skilled birth attendants in underdeveloped countries [7,12,14,15]. The aim of this review is to classify so far reported fathers’ educational needs with reference to their participation in prenatal, childbirth, and postpartum care.

## 2. Materials and Methods

### 2.1. Literature Search Strategy

The current systematic review followed PRISMA standards. Initially, a PECO (Participants, Exposure, Comparison, Outcome) review question was developed—‘’How does male partner involvement in Antenatal Education affect outcomes for both partners?’’ Methodical research of the literature written in English, referring to men’s antenatal education, was conducted between December 2021 and February 2022. All qualitative and quantitative study designs were eligible for inclusion; therefore, there were no restrictions on the design of the study, including cross-sectional, case-control, cohort, or randomized control trials. In total, 87 studies were assessed via the Google search engine (Scholarly articles), with activated passwords of the Connection Service by a Virtual Private Network (VPN), which allows access to the electronic libraries of the University of West Attica, including all international Journals’ electronic databases (HEAL-Link publishers—https://www.heal-link.gr/en/e-journals-by-publisher/ accessed on 1 December 2021). A preliminary screening was conducted using the titles and abstracts of articles retrieved from electronic databases. The search of included studies content was conducted by using the following keywords related to the review subject: “educational needs”, “antenatal education”, “prenatal care”, “perinatal care,” or just “education”, “fathers,” and “men”. All quantitative and quantitative study designs, cohort studies, randomized control trials, experimental studies, and descriptive cross-sectional studies were eligible for inclusion.

### 2.2. Inclusion and Exclusion Criteria

In this systematic review, all types of studies were accepted relevant to the theme and published from 1997 to 2021 in order to collect most of the available information about the review topic. Exclusion criteria were: (1) inability to access the full article text and (2) studies that covered antenatal education matters but not in terms of men’s educational needs. These criteria were used in order to collect studies that contain factual information about the educational needs, feelings, thoughts, and barriers that fathers face during the pregnancy and postpartum, related to the lack of information they face, and those they prefer to be informed about, so as to feel more supportive, included and ready for the new role. Therefore, one of the main criteria that led to the inclusion or exclusion of each study was whether it focused on those aspects of men’s needs and their involvement in antenatal education.

During the first identification and screening, 87 studies were collected. After examination, on the basis of their abstracts, 30 studies were removed as not specifically related to the review topic. Full texts were retrieved for the remaining 47 articles and comprehensively examined based on their findings. Excluding the next 30 papers, the analysis of the remaining 17 full versions was processed. Therefore, seventeen eligible primary research were included in the review and methodically analyzed. This process is outlined in Figure 1. The extracted information consisted of authors, year of publication, study setting, study design, study population, number of participants, and study results. During this process, the authors (five) of the current study were first working independently, followed by mutual consensus on studies that meet the criteria for inclusion in the review. The included studies were independently reviewed by two authors (Z.P. & M.A.). In the event of discrepancies, they were discussed with others or resolved through arbitration. Hence, for the purpose of this review, individual studies regarding men’s involvement in prenatal and perinatal care were included, whether the participants of the studies were men, women, health professionals, or others referring to the review subject. Since in the current review, most of the studies’ designs are qualitative, the tool used to assess the quality was the CASP systematic review checklist, showing clear internal validity, high quality of overall assessment of the study, and low risk of bias.

## 3. Results

A total number of 87 articles were retrieved, but seventeen of them were relevant to the father’s educational needs in terms of their participation in perinatal care. Table 1 presents the characteristics of the included studies. Two included studies were conducted in the United Kingdom (n = 46), three in Sweden (n = 791), one in the Republic of Malawi (n = 89), one in the USA, Madison (n = 216), one in Belgium (n = 72), four in Iran (n = 1387), one in Denmark (n = 10), one in Brazil (n = 19), one in Western Australia (n = 533), one in Turkey (n = 68) and, finally, one study was conducted through the online network of communities Reddit (n = 426).

In terms of study design, there were nine qualitative research designs, one randomized controlled multicenter trial on antenatal education, two descriptive cross-sectional studies, one quasi-experimental study, one multistage consensus method, one phenomenographic method, one quota sampling method, and one repeated measures cohort study. In almost all included studies, the population was men; in some studies, there were women too, as well as health care professionals, which also only studied samples in one research, while in one study, expectant women, men, and nurses/midwives, were included and 13 Key informants (KIs) consisting out of the following: one obstetrician, one senior nurse/midwife educator, one senior practicing nurse/midwife, one policy maker, two religious leaders, one leader on behalf of the small-scale business community, two group village heads and four employers.

Data collection methods varied across the studies, including semi-structured interviews, questionnaires, focus groups, trials, and content analysis. The findings for each study are discussed individually below.

According to the findings, the men interviewed wanted to learn more about what to expect during their partner’s birth and their role as birth partners [16]. They expressed a need for more information on the physical and emotional demands of parenthood. Women perceived that professional support was beneficial to their relationships during pregnancy [17]. Moreover, one study [18] has found that men who experience antenatal FOC (Fear Of Childbirth) are more likely to experience childbirth as frightening. Additionally, another study [19] also proposed that women and men wanted similar information about antenatal care. There is an indication that perinatal education that focuses on fathers’ needs can positively influence men’s coping behavior and relationships with their partners during pregnancy [20]. Based on some results [21], fathers have diverse needs, but the information is more important than involvement or experience. Involvement in the childbearing process is also considered, but the difference between the two clearly shows that formal information needs have more priority than involvement in the process. A study’s findings [22] may help achieve gender parity among parents by including fathers in the process of parental education.

One study [23] suggests that there was an apparent lack of healthcare education and training related to fathers’ needs. The key points of male participation in perinatal care were empathy and accountability [24]. Also, additional study results [8] revealed that most men felt a strong need for information about pregnancy, childbirth, and the postnatal period, with emphasis on feeding during pregnancy, sexual health, and pregnancy risk signs. The findings of another study [25] provide perceptions into the type of worries that some fathers experience during pregnancy, which can apprise the development of father-specific resources and perinatal education, such as perinatal loss, maternal well-being, father role, feeling unprepared, genetic or chromosomal abnormalities, gender of infant, childbirth, the well-being of infant following birth, appointments and financial pressure.

With regard to the impact of the 1-year course, a study [26] appears that fathers received support, inspiration, and information that facilitated their competence as parents and made them feel more secure in their role as fathers. They felt strengthened in their sense of responsibility and awareness of their roles as fathers in their children’s lives. The first related study in the Islamic Republic of Iran [4] showed that participants were favorable toward men’s participation in perinatal care. The majority believed that education was important for both mothers and fathers and that men did not know how to help, indicating that men should be educated on these issues. More than 95% of participants agreed with perinatal care education for men, and the content most required was signs of risks during the perinatal period and mothers’ nutrition. Soltani et al. [27] highlight the importance of training men interested in perinatal care. The significance of training the men who would be participating in various perinatal care was stressed, including the topics such as physical changes that occur during pregnancy, prenatal nutrition, risks signs during pregnancy, and postpartum care for both mom and baby.

According to Sousa et al. [28], fathers have a substantial role to play in the pregnancy and delivery processes, as their presence and support promote the safety of mothers and babies, decreasing preventable risks during pregnancy and establishing a bond between father and child from an early age. Antenatal education tailored to meet the needs of both fathers and mothers, as well as early awareness and intervention, can limit the negative impacts of perinatal anxiety and depression on parenting behavior and attitudes among parents and may increase parental coping skills [29]. Lastly, according to Turan et al. [30], expectant fathers, as well as expectant mothers, can benefit from antenatal education programs that increase their knowledge, attitudes, and behaviors regarding reproductive health. Positive results were also seen in infant feeding, spousal communication, and support under the community-based program.

## 4. Discussion

This study is the first systematic review of fathers’ educational needs relating to perinatal care. All studies demonstrated in this review showed that fathers’ participation in antenatal education could be of substantial impact on pregnancy outcome, childbirth, parental role, partner relationship, and family functioning in general. Based on the results, fathers appeared to have various needs, but some studies indicate that formal information needs have more priority than involvement in the process [21] and that educational programs should be integrative and tailored to expressed needs of fathers [21,29]. Some evidence showed that, due to a lack of support from professionals, men could feel unimportant [17,23]. Furthermore, it is indicated that a community-based approach may be a more effective method of involving men than a clinic-based approach [30]. Several studies’ participants suggest that the best places for educational programs were health centers [8], hospitals [24], and homes [24,26], with evenings and weekends being the most convenient time [24,26]. Other studies show that men prefer face-to-face education, which is also recommended by experts [31]. In Mullick’s study [32], open hours on weekdays at the clinics were the preference for the training. Furthermore, in the study of Simbar and his colleagues [4] seemed that men prioritized receiving education at health centers by midwives. Also, earlier intervention may benefit fathers’ mental health more in the longer term [22,25].

The Indian and Guatemalan studies [33,34] found that physicians or midwives can provide prenatal care to men in health centers, including childbirth preparation classes, with their preference to attend these classes accompanied by their wives. The growing influence of men during pregnancy, childbirth, and postpartum should be taken into account when crafting national policies for safe maternity care in order to increase the health of mothers and newborns and to reduce mortality and morbidity during pregnancy and the postnatal period. In Tanzania, the intervention to educate pregnant women’s husbands on danger signs and pregnancy complications resulted in a significant increase in birth attendance by skilled health providers [35].

Analytically, male involvement in antenatal education was associated with better knowledge about physical and emotional demands of parenthood [16], infant development [8,16], relationship during pregnancy and communication within the couple [17,20,29,30], less fear of childbirth and more positive experience of the event [18,21], men’s coping behavior [20,29], gender parity among parents [21], men’s empathy and accountability [4], family functioning [4,19], knowledge about health of the mother and infant [4,8,28] sexual and reproductive health [24,30], decreasing preventable risks during pregnancy [24,26,27,28] getting into a father’s role [8,25], feeling prepared [8,19], mothers’ nutrition [26,27], physical changes that occur during pregnancy [27], postpartum care for both mom and baby [27], establishing a bond between father and child from an early age [30], limitation of negative impacts of perinatal anxiety and depression [29], parenting behavior [29], infant feeding [30], and HIV prevention during pregnancy [19].

Perinatal education should address psychosocial issues such as partner well-being [36], coping with loss during pregnancy, transitioning to fatherhood, financial stress, and work/family conflict. The fact that fear associated with childbirth has been linked with negative outcomes for fathers and children [37] suggests that antenatal education needs to fully address the anxieties of men about labor and childbirth. Research shows that health services could better support fathers by offering them information about parenting from a father’s perspective or by facilitating father-specific sessions as a part of routine antenatal care programs [38,39].

Generally, it has frequently been documented that men’s education has positive effects on maternal and neonatal health [20,40,41,42]. Consequently, men’s education appears to be essential for their own adaptation to fatherhood. There are antenatal education classes for parents in many countries; in Scandinavian countries, 95% of fathers participate in such classes [43]. Moreover, it has been suggested that knowledge and attitude of both genders towards male participation in reproductive health should be addressed before marriage [1]. Educating and engaging men about seeking care for themselves and their children in order to address reproductive health needs may motivate them to participate in maternity care [44].

Health professionals encouraging behavior, according to the reported studies, is of basic importance for fathers’ sense of security [45,46], emphasizing the need for parenting courses to be led by professionals [47]. Those family courses make fathers feel less sidelined from their parental roles. Additionally, gender-synchronized activities might help in partner interaction [48]. Being in the same parenting group but also working separately in “men’s” and “women’s” groups could encourage dialogue and focus on both genders and the couple’s relationship.

The limitation of this study is that it includes only articles written in English. It is likely that such inclusion criteria have resulted in the loss of valuable information that could have been used for the review.

## 5. Conclusions

According to the current analysis, fathers desire to contribute to their child’s development by getting involved in pregnancy so they feel involved and prepared. It is emphasized that midwives’ relevant education provided to fathers in prenatal care leads to positive outcomes such as healthier relationships with their partners, earlier initiation of prenatal care, reinforcement of confidence and parents’ sense of belonging to their children, adoption, and retention of health behaviors, getting a better understanding of delivery-related complications and pregnancy complications by men, reducing men’s worries, and providing neonatal care and family function, among others. Future fathers from different countries around the world should be included in further research to differentiate cultural and regional antenatal education needs.

## Figures and Tables

**Figure 1 healthcare-11-00200-f001:**
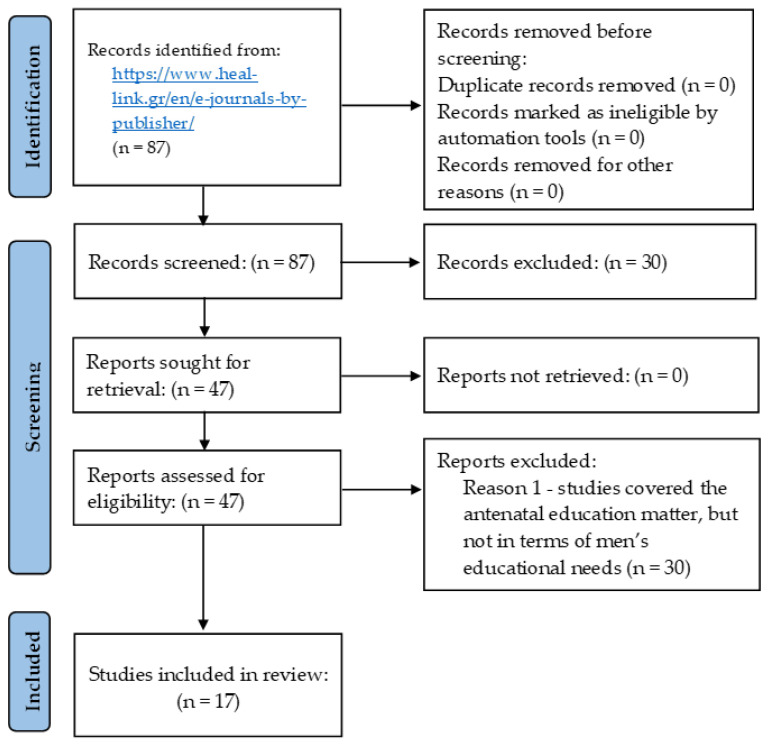
Flow diagram illustrating the article filtering process.

**Table 1 healthcare-11-00200-t001:** Characteristics of included studies.

Reference	Aim	Country	Year	Study Design	Measures	Final Sample Size	Outcomes/Conclusions
Baldwin, S.; et al. (2019). [16]	To explore men’s own perceived needs and how they would like to be supported during and beyond their partner’s pregnancy.	United Kingdom	2019	Maximum variation sampling	Face-to-face in-depth interviews	21 men	The interviewed men desired to be appropriately informed about labor and their role as birth partners. A greater understanding of the physical and emotional demands of parenthood during the early days and weeks after birth was expressed.
Bäckström, C.; et al. (2017). [17]	To explore pregnant women’s partners’ perceptions of professional support during pregnancy	Sweden	2017	Qualitative research design	Semi-structured interviews	14 men	A positive impact on the couple’s relationship was perceived by men during pregnancy as a result of professional support. Furthermore, they believed that lack of professional support would contribute to feelings of unimportance, potentially affecting mothers and babies negatively.
Bergström, M.; et al. (2013). [18]	To explore if antenatal fear of childbirth in men affects their experience of the birth event and if this experience is associated with the type of childbirth preparation.	Sweden	2013	Randomized controlled multicenter trial on antenatal education	Wijma Delivery Expectancy Questionnaire, W-DEQ A (Wijma, Wijma, & Zar, 1998)	762 men	The chance of experiencing childbirth as frightening is higher for men suffering from antenatal FOC (Fear Of Childbirth). In addition to childbirth preparation, antenatal education may help men to have a more positive experience of childbirth.
Chikalipo, M.C.; Chirwa, E.M.; Muula, A.S. (2018). [19]	To gain a deeper understanding of the education content for couples during antenatal education sessions in Malawi.	Republic of Malawi	2018	Exploratory cross-sectional descriptive study (qualitative approach)	In-depth interviews	34 women, 35 men, 7 midwives & 13 Key informants	Men and women expressed relatively similar needs referring to antenatal education, such as pregnant woman care, childbirth, child care, and family planning. Furthermore, sex and men’s roles during the perinatal periods, prevention of mother-to-child HIV transmission, and family life were the topics desired by men, in comparison to birth preparedness was more desired by women.
Diemer, G.A. (1997). [20]	To compare the effects of father-focused discussion in perinatal classes with traditional childbirth classes on expectant fathers’ stress/psychological symptom status, coping strategies, social support, and spousal relations.	USA, Madison	1997	Quasi- experimental study	1. Brief Symptom Inventory, BSI (Deroogatis & Spencer, 1982) 2. Coping Measure Scale, CMS (Lazarus & Folkman 1984) 3. Social Network Support Scale, SNSS (Fischer, 1982) 4. Supportive Behavior Questionnaire, SBQ (Wapner, 1976, adapted by Diemer, 1981) 5. Conflict Tactics Scale, CTS (Straus, 1979)	108 couples	Antenatal education for fathers can positively influence men’s coping behavior and relationships with their partners during pregnancy. Expectant fathers may benefit from small-group discussions so as to express their concerns and feelings and establish relationships with each other, which would positively affect the relationship between the couple.
Eggermont, K.; et al. (2017). [21]	To identify fathers’ needs during the labor and childbirth process.	Belgium	2017	Multistage consensus method	The questionnaire designed for this study consisted of six parts: (1) preparation for childbirth, (2) general information, (3) support from the midwives, (4) experiences of labor and childbirth, (5) needs during labor and childbirth, and (6) demographic characteristics	72 men	Information needs were more important to men than experience or involvement. Apparently, the need for information also implies a degree of involvement. A separation in needs clearly indicates that formal information needs are more relevant than being involved in the childbearing process.
Erlandsson, K.; Häggström-Nordin, E. (2010). [22]	To capture fathers’ conceptions of parental education topics, illuminated by their experiences as primary caregivers of their child immediately following birth.	Sweden	2010	Phenomeno-graphic method	In-depth interviews	15 men	Fathers should be involved in parental education in order to achieve parity with mothers in their role as parents. Among the most important topics discussed are the mother-infant separation effects on mothers, fathers, and newborns.
Mayers, A.; et al. (2020). [23]	To investigate fathers’ experience of support provided to fathers.	United Kingdom	2020	Qualitative study	Online Questionnaire, with open questions regarding the father’s emotional well-being and the offered support	25 men	Fathers perceived a lack of healthcare education and training regarding their educational needs in relation to perinatal care. In this study, they reflected that earlier intervention might have a beneficial impact on their mental health in the longer term.
Mehran, N.; et al. (2020). [24]	To explain the concept of spouse participation in perinatal care.	Iran	2020	Qualitative study	Semi-structured in-depth interviews	7 men	Empathy and accountability were the most important aspects of men’s involvement in perinatal care. As a general result, the concept of men’s participation in perinatal care has been defined as a set of accountable behaviors towards their partners based on emotional and cognitive responses, position management, support, and compassion. Improvement of the family function and mother and baby health is the favorable consequences.
Nasiri, S.; et al. (2019). [8]	To identify men’s educational needs for participation in prenatal, childbirth, and postnatal care.	Iran	2019	Descriptive cross-sectional study (cluster sampling)	Questionnaire designed based on Mortazavi and Simbar’s studies that included demographic characteristics of the subjects, their educational needs in terms of the content of the training program, the training method, the trainer, time and place of training	280 men	Nutrition, sexual health, and warning signs during pregnancy were the most important educational needs articulated by men, preferably receiving information from a physician. According to them, the best educational place is home, the health center, and the hospital, respectively; evenings hours and holidays seemed to be chosen as the best time for it.
Pilkington, P.D.; Rominov, H. (2017). [25]	To obtain insights into fathers’ worries during pregnancy by analyzing the content of posts on the Internet forum Reddit.	Online network of communities	2017	Qualitative Content Analysis of Reddit	Analyzing the content of posts on the Internet forum Reddit	426 unique users submitted the 535 posts in the final data set	The findings provide insights into the type of worries that some fathers experience during pregnancy, which can inform the development of father-specific resources and perinatal education, such as perinatal loss, maternal well-being, father role, feeling unprepared, genetic or chromosomal abnormalities, gender of the infant, childbirth, the well-being of infant following birth, appointments and financial pressure.
Reinicke, K. (2020). [26]	To explore the extent to which parenting courses attended by both the mother and the father constitute an appealing institutional service for first-time fathers and whether they find them useful in tackling the challenges they face during pregnancy and after birth.	Denmark		Qualitative study	Individual semi-structured interviews, group interview, and observations	10 men	Fathers who participated in the 1-year course and received support, inspiration, and information, experienced competence improvement regarding their parenting roles, as well as a sense of responsibility and awareness of their role as a father.
Simbar, M.; et al. (2012). [4]	To assess the educational needs of men for their participation in perinatal care.	Iran	2011	Quota sampling method	Focus group discussions	24 women & 22 men	More than 95% of participants agreed with perinatal care education for men, and the content most required was “signs of risks during the perinatal period” and “mothers’ nutrition”. The majority of participants preferred the face-to-face couples’ counseling method, at home as the best place, and evenings and weekends as the best time.
Soltani, F.; et al. (2018). [27]	To investigate men’s knowledge and attitude about participation in their wives’ perinatal care	Iran	2018	Descriptive cross-sectional study	“Men’s knowledge and attitudes about participation in perinatal care” questionnaire, designed by the research team	300 men	Among various aspects of perinatal care, the highest number of men appreciate a good level of knowledge related to the field of delivery and breastfeeding.
Sousa, B.; et al. (2021). [28]	To identify the meanings assigned by primary health care professionals to male prenatal care.	Brazil	2021	Descriptive study (qualitative approach)	Semi-structured interviews	19 primary health care professionals	The role of fathers in the pregnancy and delivery procedures can highly benefit this process, as their presence and support promote the safety of mothers and babies, decreasing preventable risks during pregnancy and establishing an early bond between father and child.
Tohotoa, J.; et al. (2012). [29]	To identify the impact of a father-inclusive intervention on perinatal anxiety and depression	Western Australia	2012	Repeated measures cohort study	A baseline questionnaire that included demographic data of age, marital status, nationality, income, and educational level, plus the Hospital Anxiety and Depression Scale	533 men (289 in the intervention group & 244 in the control group)	Improved antenatal education to meet the needs of both mothers and fathers and early awareness and intervention may limit the negative impact of perinatal anxiety and depression on parenting attitudes and behavior and increase coping skills.
Turan, J.M.; et al. (2001). [30]	To investigate methods for including men in antenatal education in Istanbul, Turkey.	Turkey	2001	Formative study (qualitative research methods)	Three studies investigating methods	30 men & 38 women	Antenatal education can have positive effects on reproductive health knowledge, attitudes, and behaviors. In the community-based program, positive effects were also seen in the areas of infant health and feeding, spousal communication, and support. It seems likely that the more intensive, continuous, and ‘support group’ nature of the community-based program for expectant fathers, compared to the clinic-based program, may be a more successful method for involving men.

## Data Availability

Not applicable.

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
