# Peer review of "Fathers’ Educational Needs Assessment in Relation to Their Participation in Perinatal Care: A Systematic Review"

_healthcare, 2023, doi:10.3390/healthcare11020200_

Round 1
Reviewer 1 Report
Dear Editor, dear authors,
I am happy for a chance to review this article. Fathers' help during the pregnancy and their role in later raising children could be of significant importance. Every effort to enable them to find their parts in these steps is substantial. That makes this article of great importance.
Introduction
The first sentence includes some numbers/references which are not properly stated:
"One-half of the active population consists of men, who make decisions about health, 26 education, economic activities of their partners, and about family matters in general. [1-27 2]."
Actually, in the whole introduction section are some irrational numbers in sentences. This also refers to discussion section.
Line 45: "…as well as attendance in some contexts" – in which?
Methods
Why in this period "from 1997 to 2021"? Did you select all studies ever published?
Figure 1 should be in line with PRISMA guidelines. If you have all elements according to PRISMA, please edit the method section accordingly. The whole method section should be described in more details according to this guideline.
Which database did you search? It has to be mentioned.
“Analysis of the risk of bias” and a “minimum standard checklist for evaluation outcomes” must be described and included (at least as appendix).
Results
Table 1 should include the paper's aims. You mentioned that you had studies focusing on factual information: educational needs, feelings, thoughts, and barriers. It would be helpful to add specific targeted outcomes of the studies in Table 1. In line with the aims, the results should be more precise.
You have to include study methodology, instruments used in studies, primary results, and general conclusions.
You have included studies that had both qualitative and quantitative study designs. I suggest focusing on one study type. Then, you can make more precise conclusions.
Discussion and conclusion
These sections are decent and appropriate to presented results.
Author Response
Please, see the attachment file.

Reviewer 2 Report
First of all, I want to note that it has been a pleasure review your manuscript. I think this is an interesting work on the needs for men's antenatal education.
After reading in depth the manuscript, I would like to make some comments
- It is not clear to me that it is a systematic review. Has it been registered in any database such as PROSPERO?
- page 1, line 24, number 26 seems to be left out....
- page 1, line 25, the references are misspelled.
- Line 35, the number 37 is left out. I think there is a mistake of having left the line number in the left margin. Revise the document, because in lines 37, 38 and 39 there are numbers that should not be there.
-Line 33 the reference must precede the full stop.
- Line 28, there is also a fault because there is a number that divides the word intervention. Please check the whole document. This should have been checked before sending the document to the journal.
- The ideal way to conduct a systematic review is to follow the PRISMA standards, (Preferred Reporting Items for Systematic Reviews and Meta-Analyses) -
- I propose to incorporate a table with the main search equations performed in the different databases.
- If it is to be a systematic review, a flow chart based on the PRISMA standards should be drawn up. It might be necessary to assess whether it is better to do a narrative review.
- Table 1: it would be convenient to put in the first column, the authors and not only the number of the reference.
- No methodological assessment of the studies has been made.
- In the discussion section, the limitations of the study remain to be discussed.
The discussion could be better structured to make its different parts clearer to the reader.
- In references section there are some DOIs that are underlined and should not be.
Author Response
Please, see the attachment file.

Round 2
Reviewer 1 Report
Dear authors,
I have a few more suggestions:
Line 43-45: The sentence: „When expectant fathers are involved in antenatal education, health behavior during pregnancy can be improved [12] and women’s use of skilled childbirth and postpartum care can be increased [13], as well pre- and post-seminar attendance in the antenatal clinics“ doesn't have sense. Please rewrite this sentence.
You have mentioned: „In total, 87 studies were assessed via Google search engine, with activated pass words of the Connection Service by a Virtual Private Network (VPN), which allows the access to the electronic libraries of the National University of Athens, including all international Journals’ electronic databases (HEAL-Link publishers - https://www.heallink.gr/en/e-journals-by-publisher/)“
Why did you use google for study search and not PubMed, Web of SCience, Scopus? Google is not an appropriate platform for a literature review.
Best regards!
